# Expression of Genes Involved in ABA and Auxin Metabolism and *LEA* Gene during Embryogenesis in Hemp

**DOI:** 10.3390/plants11212995

**Published:** 2022-11-07

**Authors:** Daniel Král, Josef Baltazar Šenkyřík, Vladan Ondřej

**Affiliations:** Department of Botany, Faculty of Science, Palacký University Olomouc, 783 71 Olomouc, Czech Republic

**Keywords:** *Cannabis sativa*, embryogenesis, auxin, abscisic acid, *LEA* gene, embryo cultures, RT-qPCR, gene expression

## Abstract

The level of phytohormones such as abscisic acid (ABA) and auxins (Aux) changes dynamically during embryogenesis. Knowledge of the transcriptional activity of the genes of their metabolic pathways is essential for a deeper understanding of embryogenesis itself; however, it could also help breeding programs of important plants, such as *Cannabis sativa*, attractive for the pharmaceutical, textile, cosmetic, and food industries. This work aimed to find out how genes of metabolic pathways of Aux (*IAA-1*, *IAA-2*, *X15-1*, *X15-2*) and ABA (*PP2C-1*) alongside one member of the *LEA* gene family (*CanLea34*) are expressed in embryos depending on the developmental stage and the embryo cultivation in vitro. Walking stick (WS) and mature (M) cultivated and uncultivated embryos of *C. sativa* cultivars ‘KC Dora’ and ‘USO 31’ were analyzed. The RT-qPCR results indicated that for the development of immature (VH) embryos, the genes (*IAA-1*, *IAA-2*) are likely to be fundamental. Only an increased expression of the *CanLea34* gene was characteristic of the fully maturated (M) embryos. In addition, this feature was significantly increased by cultivation. In conclusion, the cultivation led to the upsurge of expression of all studied genes.

## 1. Introduction

Embryogenesis refers to the process by which a multicellular embryo develops from a zygote. It is a period of intense growth, cell differentiation, and morphogenesis. In its course, the basic body plan of the plant is determined, and the formation of the apical-basal axis determines the polarity of the embryo. The result of embryogenesis is a mature embryo, which already shows the organization of the plant body and species-specific morphology in the radial grouping of stem cells in the tissue layers. Therefore, all cell divisions must be specifically organized and controlled by a series of signaling molecules and phytohormones [1,2].

The phytohormone Aux is one of the most important regulators of plant embryogenesis. It has been proven that the influence of Aux is essential for asymmetric division, the triggering of position-specific genetic programs, and, therefore, for cell differentiation not only within the framework of the development of the embryo but also of the whole plant [2,3,4,5]. The relatively short but highly versatile Aux signaling pathway (extensively reviewed in [6,7]) enables fast switching between repression and activation of gene transcription through auxin-dependent degradation of transcriptional repressors [4].

Another important phytohormone is abscisic acid (ABA). ABA is essential not only in the period of late embryogenesis by inducing a state of dormancy and inhibiting germination events [8] but is also used in many other processes of plant life. ABA affects fruit ripening and senescence, controls stomatal activity, induces senescence and leaf fall, inhibits shoot growth, stimulates the storage of proteins in seeds, and is also involved in plant adaptation to abiotic stresses such as drought, cold, or high salinity [9,10,11]. Two independent teams made the first step toward understanding the ABA signaling pathway. Thus, ABA receptors were identified [12,13]. The protein phosphatase 2C (*PP2C*) family serves as a coreceptor and negative regulator of ABA signaling, and protein kinases in the SNF1-related protein kinase 2 (*SnRK2*) family have the function of positive regulators [14,15,16]. ABA induces the expression of many genes, including the late embryogenesis abundant (*LEA*) protein gene family. LEA proteins protect cells from adverse conditions (dehydration, salinity, temperature), and their accumulation is typical in the late maturation stage of seed development [17,18].

ABA and Aux metabolism is also influenced by transcription factors belonging to the B3 DNA binding superfamily. This plant-specific B3 superfamily includes members of the LAV, ARF, RAV, and REM families [19]. Members of the LEC2-ABI3 subgroup of the LAV family, such as the well-studied *LEC2* [20], *ABI3* [21], and *FUSCA3* [22] genes, are crucial for proper seed development. These genes are essential regulators of embryo development. *LEC2* regulates the expression of genes involved in Aux [23] and storage compounds [24,25] biosynthesis in seeds. Together with *FUSCA3*, *LEC2* has a role in determining cell identity during embryogenesis [26,27]. *ABI3* is essential for seed storage and inducing dormancy and water loss tolerance [28,29]. This gene could also be involved in responses to abiotic stress conditions [29].

Gibberellins (GAs) are involved in many developmental processes during the plant life cycle. Generally, GA is perceived as a compound that promotes plant growth and development and as a functional antagonist to ABA [30]. GA causes axis elongation during early embryogenesis [31] and is essential for the production of viable seeds [32], promotes stem and leaf growth, and is also involved in flower and fruit development [33]. The GA signaling pathway is triggered by the binding of GA to the GID1 receptor, which can trigger the degradation of repressors of GA signaling, the DELLA proteins [34,35]. As has been elucidated, DELLA proteins inhibit the activity of transcription factors such as the LEC1 protein. LEC1 initiates the transcription of many genes essential for embryogenesis; therefore, the role of GA is not negligible even in the late stages of embryo development [36].

Cytokinins (CKs) have been known to be promoting factors of cell division since their discovery [37]. Later, it was found that they substantially influence many growth and developmental stages. For example, CKs are involved in seed, shoot, and root development, germination, senescence [38], nodulation [39], and circadian rhythm [40]. CKs have been shown to interact significantly with Aux and often antagonistically [41]. For example, during the early stages of embryogenesis, Aux counteracts CK signaling [42], and during root meristem establishment, CKs suppress Aux signaling and transport [43]. Among the members of the CK signaling pathway, members of the Cytokinin Response Factors (CRFs) family might be important for embryogenesis. The expression of CRFs is not only regulated by CKs but also by Aux and ABA [44,45]. The importance of CRFs together with auxin-related PIN proteins has been demonstrated in these studies [46,47].

*Cannabis sativa* L. (hemp) is an annual and naturally dioecious plant belonging to the family of *Cannabaceae* [48,49]. The separation of this genus from its sister genus *Humulus* occurred in the Tibetan Plateau region 27.8 million years ago [50]. It is one of the first domesticated plants, and people have been discovering its vast potential for use for more than 10,000 years [51,52,53]. Currently, it attracts attention mainly for its content of many biologically active substances and fibers, good cultivation properties [54,55], and its industrial [56,57], ornamental [58], and pharmaceutical [59] applications. Therefore, hemp is attractive for genetic engineering and biotechnology [60,61]. Genome sequencing was an essential point for molecular studies of this plant [62,63]. The first transformation system for hemp was already introduced in 2000 [64], but in the following years, it turned out that hemp is relatively resilient to transformation [65,66]. However, in vitro cultivation techniques are already well-known and established [67,68]. Like in vitro seed germination, the culture of isolated embryos could be a potential alternative to the traditional procedure of vegetative hemp propagation. The process of obtaining sterile plant material identifying elite genotypes would thus be simplified and made more efficient by promoting biotechnologies using seedling-derived tissues [69].

This work aimed to determine the expression profiles of the genes *IAA-1*, *IAA-2*, *X15-1*, *X15-2*, *PP2C-1*, and *CanLea34* in embryos of two *C. sativa* cultivars ‘KC Dora’ and ‘USO 31’ depending on the developmental stage and embryo cultivation in vitro. This knowledge supports a deeper understanding of plant embryogenesis in vivo and in vitro. In addition, it could be helpful for breeding programs and hemp biotechnology.

The investigated genes are expressed differently in response to drought stress [70]. To our knowledge, no study has yet been conducted on these genes in relation to embryogenesis in *Cannabis*. Understanding the mechanisms of this plant’s tolerance to drought stress is essential, as drought is one of the most fundamental problems to which *Cannabis* cultivation is exposed. For example, as one of the largest growers of hemp, China is facing increasing water scarcity, reflected in the reduction of production and quality of hemp materials [71,72,73].

## 2. Results

Changes in gene expression during embryogenesis of two *Cannabis sativa* cultivars, ‘KC Dora’ and ‘USO 31’, were measured by RT-qPCR. The cDNA obtained by transcription from RNA isolated from embryos at the walking stick (Ws e) and mature (M e) stages of development and from their embryo cultures was analyzed. The genes involved in Aux metabolism (*IAA-1*, *IAA-2*, *X15-1*, and *X15-2*) were studied, as well as the *PP2C-1* gene, which is involved in ABA metabolism alongside the *CanLea34* gene, which belongs to the *LEA* gene family related to plant embryogenesis.

### 2.1. Establishment of Embryo Cultures

Embryos were isolated from the seeds of two *Cannabis* cultivars in a sterile environment, and those at the walking stick and matured stages were used to establish in vitro embryo cultures (Figure 1). No genotype-dependent morphological difference in embryos was observed at day 0 of cultivation. After seven days of cultivation, the success of embryo culture establishment was evaluated. 22% of ‘USO 31’ and 14.3% of ‘KC Dora’ embryos were aborted or severely deformed. This difference could indicate that the choice of genotype is essential for successful embryo cultivation. In addition, the results show that the stage of embryo development plays an even greater role. 71.4% of cultivated WS e and 92.6% of cultivated M e were viable.

### 2.2. Expression Patterns in Walking Stick Embryos

Results show that only two genes, *IAA-1* and *IAA-2*, were expressed at this stage of development. *IAA-1* was expressed 2.15-fold more (‘USO 31’) and up to 3.9-fold more (‘KC Dora’) compared to the control. While *IAA-2* was expressed 0.87-fold less in WS e (‘KC Dora’), WS e (‘USO 31’) expressed this gene up to 2.86-fold more. Only this gene showed a significant difference between genotypes. The product of the *IAA-1* gene is involved in Aux catabolism, while *IAA-2* encodes an Aux-induced protein. The other Aux-induced protein genes *X15-1* and *X15-2* were hardly expressed at this stage. Deficient expression of *PP2C-1* was also detected. Therefore, it can be concluded that for embryos at the walking stick stage, only *IAA-1* and *IAA-2* are transcribed among all these investigated genes (Figure 2).

### 2.3. Expression Patterns in Mature Embryos

The obtained expression profiles of all these studied genes were highly variable (Figure 2). In M e of ‘USO 31’, significant expression of *IAA-1*, *IAA-2*, and *X15-1* genes were measured. Compared to WS e, *IAA-1* was expressed up to 4.6-fold more in this cultivar, *IAA-2* 2.1-fold more, and the most significant change was detected in *X15-1*, which was up to 508-fold more. In ‘USO 31’ M e, *X15-1*, and *PP2C-1* genes were slightly up-regulated compared to the control. M e of the second cultivar did not significantly express any of the genes examined, and the expression of *IAA-1* was up to 5.1-fold lower and *IAA-2* up to 8.7-fold lower compared to WS e. The results suggest that these expression differences could be due to differences in genotypes. However, we consider that although the embryos were morphologically identical, M e of ‘KC Dora’ was already in a fully dormant state and thus were transcriptionally inactive. The high expression measured in M e of ‘USO 31’ indicates that the seeds used were still metabolically active and, therefore, only in the preparation stage for dormancy.

### 2.4. Gene Expression Affected by Cultivation

Cultivated embryos at the walking stick (WS ec) and matured (M ec) stages were analyzed for changes in expression levels due to cultivation (Figure 3). For all genes examined, it was found that cultivation triggers and up-regulates their expression independent of the developmental stage. The most significant difference was observed for the *X15-2* and *PP2C-1* genes compared to uncultivated embryos. In the WS ec, *X15-2* was expressed up to 601-fold more (‘USO 31’), 95.1-fold more (‘KC Dora’) and *PP2C-1* 84.5-fold more (‘USO 31’), and 131.3-fold more (‘KC Dora’). The same comparison showed that M ec expressed *X15-2* 4. 7-fold more (‘USO 31’), 88.3-fold more (‘KC Dora’) and *PP2C-1* 3.4-fold more (‘USO 31’), and 53-fold more (‘KC Dora’). These genes, together with *IAA-1*, were the most highly transcribed in embryo cultures compared to other samples. *IAA-1* expression was increased 3.9-fold (WS ec of ‘USO 31’), 1.3-fold (WS ec of ‘KC Dora’), 1.04-fold (M ec of ‘USO 31’), and up to 7.7-fold (M ec of ‘KC Dora’) by cultivation. It can be concluded that in vitro cultivation of embryos leads to activation of both Aux and ABA signaling pathways.

### 2.5. Expression Pattern of CanLea34

*CanLea34* was the most expressed of all genes studied in all samples (Figure 4). The results show that this gene is actively transcribed already during the walking stick developmental stage. However, it was confirmed that expression would be higher in mature embryos. Of all the studied genes, transcription of this gene was the most up-regulated by cultivation. The lowest expression was found in WS e, but even this is 135-fold (‘USO 31’) or 155-fold (‘KC Dora’) higher than in the control variant. M e expressed this gene even 20.7-fold (‘USO 31’) and up to 30-fold (‘KC Dora’) more. However, the most significant change in expression was observed in embryo cultures. By cultivation of WS e of ‘USO 31’ expression was increased up to 225-fold, WS e of ‘KC Dora’ 95-fold, M e of ‘USO 31’ 5-fold, and M e of ‘KC Dora’ 6.2-fold.

Due to too significant differences in *CanLea34* gene expression, a plot with logarithmic plotting of the data was used for a more explicit graphical representation.

## 3. Discussion

Embryo culture techniques are among the oldest and most successful in vitro procedures. They have been used to deeply understand plant embryogenesis for vegetative propagation and to obtain viable hybrid embryos after interspecific or intergeneric crosses, which cannot be developed in vivo [69,74]. This method avoids the abortion of embryos after distant hybridization, eliminates the effect of postzygotic incompatibility, and overcomes absolute inhibition of germination. Proper embryo development in vitro is dependent on many factors. It has been found that the success rate of embryo culture establishment is strongly affected by the stage of embryo development; the more mature the embryo, the easier it is to cultivate [74]. The influence of genotype is also not negligible. As our results also show, even very close cultivars can have different cultivation success rates [75], which may be influenced by distinct expression profiles of genes essential for embryogenesis and germination.

Only *IAA-1* and *IAA-2* genes were slightly up-regulated in WS e. Aux accumulates the most in the early stages of embryogenesis [3,4,5,76]. *IAA-1,* or also *GH3.6,* encodes the enzyme indole-3-acetic acid-amido synthetase GH3.6. GH3 proteins are responsible for catalyzing the ATP-dependent formation of Aux conjugates with amino acids. In this way, Aux is reduced in the cells, and its homeostasis is restored [77]. Thus, it can be concluded that during the walking stick phase of development, the amount of cellular Aux must be regulated, which could indicate a higher concentration of Aux. This is also compatible with a higher expression of the *IAA-2* gene, which encodes the indole-3-acetic acid-induced protein ARG7.

For all genes examined, a significantly different expression profile was found between the M e of the cultivars. In dormancy, gene expression is suppressed to a minimum [78], which corresponds to the expression levels in M e of ‘KC Dora’. However, the M e of the second cultivar significantly expressed all genes examined. We do not suggest that this is due to a genotype difference. Despite being morphologically identical, the embryos used were still metabolically active with the Aux signaling pathway triggered and thus only at the stage of preparation for dormancy. The up-regulated expression of *PP2C-1*, which product serves as a coreceptor, and negative regulator of ABA signaling [15,16], indicates that the ABA signaling pathway has also been activated. ABA is most concentrated during the period of cotyledon growth. Thereafter, its amount decreases, and the next peak is during the period of pericarp development [76].

In vitro plant development is strongly influenced by the culture conditions. In particular, the composition of the culture medium, light, temperature, and humidity play an important role [79,80,81]. Stressful environments can also arise when conventional sealing using parafilm is used (used in the experiment). Restriction of gas exchange, the imbalance between CO_2_ and O_2,_ along with the accumulation of H_2_O_2,_ causes stress to cultures, leading to altered expression profiles and up-regulation of stress-related genes [82,83]. Cultivation-induced germination results in growth and cell differentiation but in a stressful environment. The significant increase in gene expression that was measured in almost all samples tells us that both Aux and ABA signaling pathways are triggered by cultivation. This result could also be influenced by the cooperation of sucrose in the culture medium with Aux, which triggers the expression of genes involved in Aux metabolism and causes an increase in endogenous ABA [76,84,85].

ABA induces the expression of *LEA* genes which products are characteristic of the late developmental stages of plant embryogenesis [17,18,74]. This was confirmed by a significant increase in *CanLea34* expression in M e. The significant difference between the genotypes of M e supports that the embryos were differentially matured. The multiplied expression confirmed that the embryos were exposed to abiotic stresses during the cultivation.

## 4. Materials and Methods

### 4.1. Cultivation of Cannabis Plants

*Cannabis sativa* cv. ‘USO 31′ and cv. ‘KC Dora’ plants served as this study’s experimental material source. The plants were grown in a greenhouse. At the time of 4 to 5 months after sowing, individual seeds were collected at various stages of maturity and used for the experiment immediately. ‘USO 31′ seeds were provided by Agritec (Šumperk, Czech Republic), and ‘KC Dora’ seeds were provided by SEMO (Smržice, Czech Republic).

### 4.2. Embryo Isolation and Establishment of Embryo Cultures

A mixed sample of seeds at different stages of maturity was collected from each *Cannabis* cultivar from multiple plants. For embryo isolation, the seed surface was sterilized using 70% ethanol for 1 min, followed by 2.5% chloramine T, and then washed three times with sterile water. The walking stick (WS) and matured embryo (M) were isolated in an aseptic environment. Representative embryos of both stages (WS and M) were placed on single-use plastic Petri dishes with a culture medium in threes. Culture medium composition was as follows: full MS medium with vitamins [86], 4.42 g·dm^−3^; sucrose, 30 g·dm^-3^; plant agar, 8 g·dm^−3^; ascorbic acid, 0.02 g·dm^−3^; indole-3-butyric acid (IBA), 0.01 g·dm^−3^; 6-benzylaminopurine (BAP), 0.01 g·dm^−3^. Cultivation was carried out in a phytotron under the following conditions 16-h light/8-h dark cycle at 22 °C at 40% relative humidity. After one week of cultivation, WS and M embryos in Petri dishes usually produce three seedlings.

### 4.3. RT-qPCR Analysis

Total RNA was isolated with a Spectrum Plant Total RNA Kit (Sigma-Aldrich, Prague, Czech Republic). Isolated RNA was treated with DNase I Amplification Grade (Sigma-Aldrich, Prague, Czech Republic) and checked for quality and absence of genomic DNA by agarose gel electrophoresis. The concentration was determined at A260/A280 ratio using a NanoDrop 2000 spectrophotometer (ThermoScientific, Prague, Czech Republic). Complementary cDNA was synthesized using a reverse transcription kit (Bioline, Prague, Czech Republic). Primers for all investigated genes were taken from the article by van Bakel et al. [62], except for the primer for the *CanLea34* gene, which was designed using Primer 3. For detailed information of all the primers used, see Appendix A. Quantitative PCR was performed using SensiFAST SYBR No-ROX Kit (Bioline, Prague, Czech Republic) in 96-well plates by CFX Connect Real-Time PCR Detection System (Bioline, Prague, Czech Republic). The expression levels of different sampling cycles were normalized with Actin (*Act*) gene (Appendix A) as a reference gene and compared against the expression level in the control sample—*Cannabis* young leaves. The specificity of the PCR products was verified by melting analysis. Relative expression was determined using a method considering the primer efficiencies according to Pfaffl [87].

### 4.4. Statistical Analyses

The obtained results were analyzed using online freeware ASTATSA. The data were first subjected to an analysis of variance, and then the means were compared using Tukey’s HSD test based on a one-way ANOVA [88]. Significant differences between the monitored parameters (Appendix A) are represented in the graphs by asterisks (*p* ≤ 0.01 “∗∗”; *p* ≤ 0.05 “∗”) and circles (*p* ≤ 0.01 “○○”; *p* ≤ 0.05 “○”). Relative expression values are presented in graphs as means ± standard deviation determined from three technical and two biological replicates.

## 5. Conclusions

Embryogenesis is a complex process that requires the proper interplay of many genes. It is known that among the phytohormones, Aux and ABA, but also GA, CK, and ethylene, play a significant role. It has been demonstrated that the success of embryo culture establishment increases with the stage of embryo development and maturity. Cultivation is a period of growth but also of significant stress, which was reflected by an increase in the expression of all genes examined. Regarding the studied genes and their relative expressions, we have found that for the development of immature embryos, the genes *IAA-1* and *IAA-2* are likely to be fundamental, and for the fully matured embryos, only the *CanLea34* gene is. The knowledge provided by this study may help to deepen our understanding of plant embryogenesis in *Cannabis*. In addition, it may be useful for breeding programs of this plant. However, further research is needed to comprehensively understand plant embryogenesis in order to investigate the exact functions of not only these genes but also others related to the metabolism of other phytohormones.

## Figures and Tables

**Figure 1 plants-11-02995-f001:**
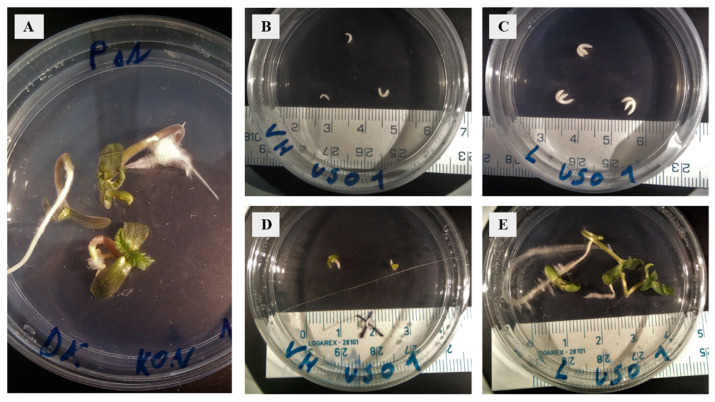
Establishment of embryo cultures and cultivation of walking stick and matured embryos of two cultivars of *Cannabis sativa*. (**A**) Mature ‘KC Dora’ embryos, day 7 of cultivation. (**B**) Walking stick ‘USO 31’ embryos, day 0 of cultivation. (**C**) Mature ‘USO 31’ embryos, day 0 of cultivation. (**D**) Walking stick ‘USO 31’ embryos, day 7 of cultivation. (**E**) Mature ‘USO 31’ embryos, day 7 of cultivation.

**Figure 2 plants-11-02995-f002:**
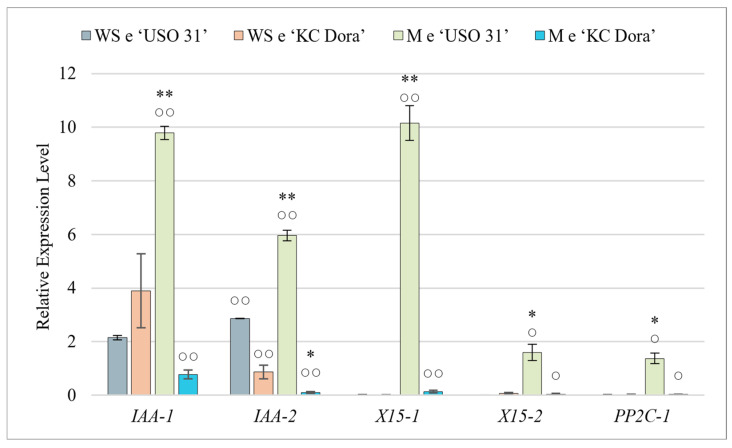
Quantification of relative expression of *IAA-1*, *IAA-2*, *X15-1*, *X15-2*, and *PP2C-1* genes in walking stick (WS e) and matured embryos (M e) of two cultivars of *Cannabis sativa* ‘USO 31’ and ‘KC Dora’. Expression data were normalized using *Act* as housekeeping gene and calibrated relative to young leaves. Significant differences in gene expression in matured embryos (Appendix A) are represented by asterisks (*p* ≤ 0.01 “∗∗”; *p* ≤ 0.05 “∗”) and between genotypes (Appendix A) by circles (*p* ≤ 0.01 “○○”; *p* ≤ 0.05 “○”).

**Figure 3 plants-11-02995-f003:**
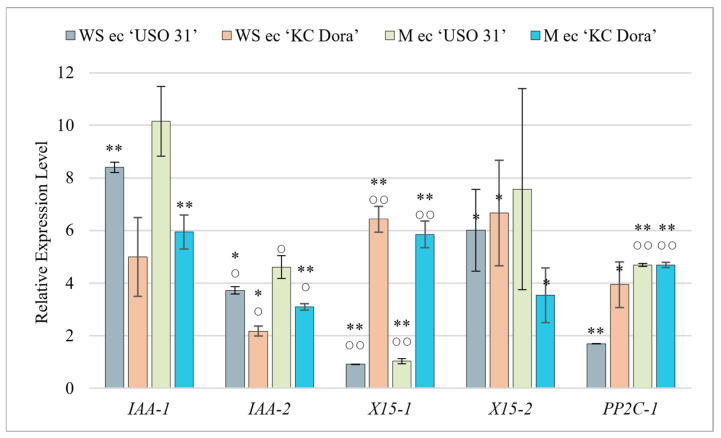
Quantification of relative expression of *IAA-1*, *IAA-2*, *X15-1*, *X15-2*, and *PP2C-1* genes in the culture of walking stick (WS ec) and matured embryos (M ec) of two cultivars of *Cannabis sativa* ‘USO 31’ and ‘KC Dora’. Expression data were normalized using *Act* as housekeeping gene and calibrated relative to young leaves. Significant differences in gene expression compared to isolated embryos (Appendix A) are represented by asterisks (*p* ≤ 0.01 “∗∗”; *p* ≤ 0.05 “∗”) and between genotypes (Appendix A) by circles (*p* ≤ 0.01 “○○”; *p* ≤ 0.05 “○”).

**Figure 4 plants-11-02995-f004:**
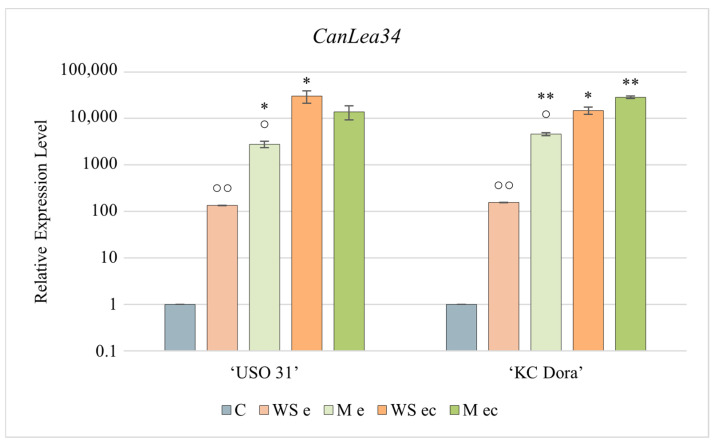
Expression profile of the *CanLea34* gene corresponding to the developmental stage and cultivation of *Cannabis sativa* ‘USO 31’ and ‘KC Dora’ embryos. Using RT-qPCR, the relative expression level was determined in walking stick (WS e), matured (M e), and embryo cultures based on these embryos (WS ec; M ec). Expression data were normalized using *Act* as housekeeping gene and calibrated relative to young leaves. Significant differences in gene expression in matured/cultivated embryos (Appendix A) are represented by asterisks (*p* ≤ 0.01 “∗∗”; *p* ≤ 0.05 “∗”) and between genotypes (Appendix A) by circles (*p* ≤ 0.01 “○○”; *p* ≤ 0.05 “○”).

## Data Availability

The data presented in this study are available on request from the corresponding author.

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
