# Peer review of "Expression of Genes Involved in ABA and Auxin Metabolism and LEA Gene during Embryogenesis in Hemp"

_plants, 2022, doi:10.3390/plants11212995_

Round 1
Reviewer 1 Report
In this Manuscript, Král et al. determined the expression profiles of the genes IAA-1, IAA-2, X15-1, X15-2, PP2C-1, and CanLea34 in embryos of two C. sativa cultivars 'KC Dora' and 'USO 31' depending on the developmental stage and embryo cultivation in vitro. Although the topic is attractive, there are some concerns that should be addressed.
-Generally, the manuscript is well organized but there are some typographical and grammatical errors.
-The paper title is well stated, it is informative and concise.
-Abstract is well structured.
-The introduction was not well written, and it is too briefly presenting the subject and research problem.
Line 55: Please provide reference(s) (suggestion: https://doi.org/10.1016/j.indcrop.2020.113026)
Line 59: “….its bioeconomic potential” should be changed to ““….its industrial (10.3906/bot-1907-15), ornamental (https://doi.org/10.3390/plants11182383), and pharmaceutical (https://doi.org/10.1007/978-981-16-8822-5_4) applications”
-Material and research methods are presented appropriately. The experimental setup and the description in the methods section are well structured, and the statistical analysis is correctly performed.
-The results obtained in this study are interesting. Results are presented correctly. However, the authors should present the images from the obtained embryos at different stages.
-In general, the discussion of the results is presented appropriately.
- Conclusion, the authors should provide a section for the conclusion. The main outcomes of the current study and future studies should be discussed in the conclusion section.
Author Response
Dear reviewer,
Thank you very much for your review and suggestions on improving our work.
- We have corrected all addressed typographical and grammatical errors and what we found.
- We have corrected the introduction and used the citations you suggested.
- We have added the image of the embryos to the manuscript, and we will add the rest of the photos to the Supplementary materials.
- We have created and provided the section
- ’Conclusion’ to the manuscript, where we discuss the current and future studies.
Kind regards,
Josef Baltazar ŠenkyÅ™ík

Reviewer 2 Report
The study carried out by Kral and colleagues is very brief. It should be considered for publication after addressing below issues.
1. Authors should expand the study as embryogenesis is not limited to few genes studied. ABA metabolism is also effected by several transcription factors more likely B3 domain transcription factors such as Leafy cotyledon 2, FUSCA3. These B3 domain TFs are crucial for embryogenesis and seedling development. For this refer to these articles
· Functional Plant Biology 2020: 48(2) DOI: 10.1071/FP19260
· Frontiers in Plant Science 2017: 8 DOI: 10.3389/fpls.2017.01604
2. Authors should consider the effect of other genes on phytohormones metabolism
3. Another important hormone gibberellin that act antagonistically to ABA should be studied along to efficiently describe the whole process of embryogenesis.
4. Authors should observe morphological changes during embryo development because studied phytohormones also effect plant physiology under stress and normal conditions.
5. Support your finding with the pictures of cultivated embryos and growing seedlings.
6. Provide list of primers for all studied genes in supplementary file.
7. Replace comma with decimal for MS media composition.
8. Italicize the specie name at line 152.
Author Response
Dear reviewer,
Thank you very much for your review and suggestions on improving our work.
- Authors should expand the study as embryogenesis is not limited to few genes studied. ABA metabolism is also effected by several transcription factors more likely B3 domain transcription factors such as Leafy cotyledon 2, FUSCA3. These B3 domain TFs are crucial for embryogenesis and seedling development. For this refer to these articles
- We have expanded the Introduction section with additional information regarding the ABA metabolism and B3 domain transcription factors (LAV, ARF, RAV, and REM families).
- We have cited the suggested articles.
- Nevertheless, the studies we cited were mostly done on soy, where the genome is well known, and the genes are annotated. These genes are unknown to us in connection with Cannabis sativa L.; for now, we cannot study the relative expression during embryogenesis. We thank you for your points for improving our work, and we will consider using them in follow-up studies when possible.
- Authors should consider the effect of other genes on phytohormones metabolism
- We have considered the effect of other genes on phytohormones metabolism, and we have added additional information in the ‘Introduction’ and ‘Conclusion’ sections.
- Another important hormone gibberellin that act antagonistically to ABA should be studied along to efficiently describe the whole process of embryogenesis.
- Thank you for this suggestion. We will be considering expanding our future study with these issues when possible.
- Authors should observe morphological changes during embryo development because studied phytohormones also effect plant physiology under stress and normal conditions.
- We expanded our manuscript with the picture of embryos. The figure shows that the morphology of the embryos and their in vitro cultures did not show any morphological abnormalities. Also, no significant morphological changes were observed between the varieties. Therefore, we did not proceed to more detailed morphological studies (e.g., microscopy, etc.). The only thing that could have influenced the morphology and hence the stress-induced change in the relative expression of the selected genes would have been contaminants, which were excluded from our analyses.
- Support your finding with the pictures of cultivated embryos and growing seedlings.
- In addition to the enclosed picture in the manuscript. If necessary we will provide the rest of the photographic documentation of experimental plant material.
- Provide list of primers for all studied genes in supplementary file.
- Thank you for reminding us. We will provide the list in a supplementary file.
- Replace comma with decimal for MS media composition.
- Italicize the specie name at line 152.
- Done
Kind regards,
Josef Baltazar ŠenkyÅ™ík

Round 2
Reviewer 1 Report
All my comments have been addressed. I think that the current version of the manuscript can be published in Plants.
Reviewer 2 Report
Accept in present form